# Detection and Clinical Associations of Autoantibodies to Heterogeneous Nuclear Ribonucleoprotein (hnRNP) A2/B1 in Patients with Systemic Sclerosis

**DOI:** 10.3390/ijms26188892

**Published:** 2025-09-12

**Authors:** Antonio Tonutti, Natasa Isailovic, Lukas Frischknecht, Francesca Motta, Minoru Satoh, Carlo Selmi, Maria De Santis, Angela Ceribelli

**Affiliations:** 1Department of Biomedical Sciences, Humanitas University, 20072 Pieve Emanuele, Milan, Italy; antonio.tonutti@humanitas.it (A.T.); francesca.motta2@humanitas.it (F.M.); carlo.selmi@hunimed.eu (C.S.); maria.de_santis@hunimed.eu (M.D.S.); 2Rheumatology and Clinical Immunology, IRCCS Humanitas Research Hospital, via Manzoni 56, 20089 Rozzano, Milan, Italy; natasa.isailovic@humanitasresearch.it (N.I.); lfrischknecht@bluewin.ch (L.F.); 3Department of Medicine, Kitakyushu Yahata-Higashi Hospital, Kitakyushu 805-8534, Japan; satohm@health.uoeh-u.ac.jp; 4Department of Human, Information and Life Sciences, University of Occupational and Environmental Health, Kitakyushu 805-8534, Japan

**Keywords:** systemic sclerosis, cancer, gastrointestinal involvement, spliceosome, immunology

## Abstract

Autoantibodies targeting heterogeneous nuclear ribonucleoproteins (hnRNPs) have been seldom described in autoimmune diseases but remain poorly characterized in systemic sclerosis (SSc). This study aims to investigate the prevalence and clinical significance of anti-hnRNP autoantibodies in SSc. Serum samples from 25 well-characterized SSc patients were analyzed using protein immunoprecipitation (IP) to detect autoantibodies against hnRNP components A1, A2/B1, C1/C2, H, L, and U. Clinical data including organ involvement and autoantibody profiles were also collected over a 10-year follow-up period. Anti-hnRNP A2/B1 autoantibodies were identified in 40% of SSc patients and significantly associated with gastrointestinal involvement (80% vs. 27%; *p* = 0.015; OR 17, 95% CI 2.2–381). Additional components such as anti-hnRNP L antibodies exhibited variable protein-IP band patterns, with a trend toward an association between a “double” band pattern and cancer history (*p* = 0.066). Anti-hnRNP U antibodies were detected in a single patient presenting with severe digital ulcers. No patient tested positive for antibodies against other components, including A1, C1/C2, and H. In this preliminary hypothesis-generating study, anti-hnRNP autoantibodies were frequent in SSc patients with distinct prevalence and clinical associations depending on the target component. Anti-hnRNP A2/B1 correlate with gastrointestinal involvement but, contrary to previous reports, show no association with arthritis. Further exploration on anti-hnRNP L and the rarer anti-hnRNP U autoantibodies is warranted.

## 1. Introduction

Antinuclear antibodies (ANAs) and their antigenic targets are a defining feature of connective tissue diseases (CTDs) including systemic sclerosis (SSc), in which anticentromere (ACA), anti-topoisomerase I (anti-TOPO1), and anti-RNA polymerase III (anti-POLR3) are considered disease-specific and incorporated into the classification criteria [1,2]. Notably, a substantial proportion of patients remain negative for all three classification criteria autoantibodies [3,4], but the autoantibody spectrum in SSc is broader. Beyond “classification criteria” antibodies, additional specificities have been described, including antibodies to nucleolar antigens [5,6], as well as others—such as anti-U1RNP and anti-Ro52—that may be observed in SSc but are not specific for the disease [7,8].

In SSc, the autoantibody profile correlates with the cutaneous subset (diffuse—dcSSc, or limited—lcSSc), risk and type of organ involvement [9,10], disease severity, as well as—according to emerging evidence—with comorbidities, primarily cancer [4,11,12]. In recent years, new antigenic targets have been reported, albeit with a seldom-limited visibility. Among them are autoantibodies directed against heterogeneous nuclear ribonucleoproteins (hnRNPs), a family of spliceosome proteins with multiple components (designated from A1 to U) that play a central role in RNA metabolism, DNA damage repair, tumorigenesis, and viral nucleic acid sensing [13]—functions that overlap with the pathways targeted by many, if not most, CTD-related autoantibodies. Anti-hnRNP antibodies have previously been reported in systemic lupus erythematosus, rheumatoid arthritis, and other CTDs, often with variable and poorly defined clinical associations, such as arthritis [14,15,16,17]. In this study, we sought to identify, quantify, and characterize the clinical significance of anti-hnRNP antibodies in a cohort of SSc patients.

## 2. Results

### 2.1. Patients’ Characteristics

Patients’ characteristics are detailed in Table 1. Twenty-five patients with SSc were consecutively enrolled: 21/25 (84%) were female, the mean age at disease onset was 53 years (±14), and the mean disease duration was 16 years (±10). Four patients (16%) had a history of cancer. According to the LeRoy subsets, dcSSc was reported in 4 cases (16%) and the median modified Rodnan skin score (mRSS) was 4 (IQR 2–8). Regarding vascular manifestations, digital ulcers were diagnosed in 11 cases (44%), telangiectasias in 15 (60%), and PAH in 2 (8%). ILD, primary heart involvement, and arthritis were each diagnosed in 5 patients (20%), calcinosis in 7 (28%), and gastrointestinal involvement in 12 (48%). No episodes of scleroderma renal crisis or myositis were recorded during follow-up.

All patients were ANA-positive on HEp-2 IIF. In 20/25 cases (80%), a single ICAP pattern was identified (https://anapatterns.org, accessed on 10 August 2025): centromere (AC-3) in 11 cases (44% of the cohort), fine speckled (AC-4) and clumpy nucleolar (AC-9) in 4 cases each (16%), and nuclear dots (AC-6/7) in 1 case (4%). Five patients showed two co-existing ANA patterns on HEp-2 IIF: centromere (AC-3) in 4 cases (16%), associated with cytoplasmic reticular/AMA (AC-21) in 2 cases, nuclear homogeneous (AC-1) in 1 case, and dense fine speckled (AC-2) in 1 case; the fifth patient had coexisting clumpy nucleolar (AC-9) and fine speckled (AC-4) patterns.

SSc-specific autoantibodies included ACA in 15 patients (60%), anti-TOPO1 in 4 (16%), and anti-POLR3 in 3 (12%).

### 2.2. Detection of Autoantibodies Towards hnRNP Antigens

Protein-IP revealed anti-hnRNP autoantibodies directed towards different components (Figure 1A). Anti-hnRNP A2/B1 were detected in 10 patients (40%), while anti-hnRNP U was found in only one (4%) patient. No patients tested positive for anti-hnRNP A1, C1/C3, or H.

Based on previously reported clinical relevance and the observed distribution in our cohort, further analyses were performed according to the presence of anti-hnRNP A2/B1 confirmed by IP-WB (Figure 1B).

### 2.3. Characteristics of SSc Patients with Anti-hnRNP A2/B1

Table 1 summarizes the demographic, clinical, and serological characteristics of patients positive versus negative for anti-hnRNP A2/B1. Overall, the demographic features were similar between groups, although the anti-hnRNP A2/B1-positive patients had a numerically higher proportion of males (30% vs. 7%) and a lower prevalence of cancer history (0% vs. 27%), without statistical significance. The distribution of clinical domains was comparable, but dcSSc was more frequent among anti-hnRNP A2/B1-positive patients (30% vs. 7%), without significance. Gastrointestinal symptoms were significantly more frequent in this group (80% vs. 27%; *p* = 0.015). The distribution of SSc-specific autoantibodies was similar, and the overall mortality did not differ between groups. On multivariable logistic regression, anti-hnRNP A2/B1 positivity was significantly associated with gastrointestinal involvement (OR 17; 95% CI 2.2–381; *p* = 0.019), independent of sex and disease duration.

The distribution of ANA patterns on HEp-2 IIF was similar in patients with and without anti-hnRNP A2/B1, but there was a trend towards more nucleolar patterns in patients without anti-hnRNP A2/B1 (0% vs. 33%; *p* = 0.06). Briefly, the other patterns encompassed: centromere (AC-3) in 6 (60%) positive and 9 (60%) negative patients; fine speckled (AC-4) in 3 (30%) positive and 2 (13%) negative patients; AMA-like (AC-21) in 0 positive and 2 (13%) patients; nuclear dots (AC-6/7) in 1 positive and 0 negative patients; and homogeneous (AC-1) and dense fine speckled (AC-2) in 0 positive and 1 (7%) negative patients each.

Given the high prevalence of ACA and lcSSc in this cohort, subgroup analyses were performed within these subsets to assess the clinical relevance of concomitant anti-hnRNP A2/B1 positivity (Table 1). Both comparisons showed no significant differences, apart from the consistent association between anti-hnRNP A2/B1 and gastrointestinal involvement.

Even if not shown in the results section, when comparing patients according to anti-hnRNP L status, no significant differences were observed between those negative and positive for one or two bands on protein-IP. Despite the small numbers, a trend towards significance was seen between the band pattern and cancer history: among cancer cases (*n* = 4), 2 (50%) had “double” hnRNP L status versus 2/21 (9.5%) in cancer-free patients, whereas “single” hnRNP L was seen in 1/4 (25%) cancer cases versus 15/21 (71.4%) cancer-free patients (*p* = 0.066).

Anti-hnRNP U autoantibodies were reported in a single patient: a woman with ACA-positive lcSSc and digital ulcers requiring multiple vasoactive treatments.

## 3. Discussion

To date, few studies have investigated the presence and clinical relevance of anti-hnRNP autoantibodies in patients with systemic sclerosis (SSc). Previous research has primarily focused on antibodies targeting the A2/B1 component, extensively characterized in rheumatoid arthritis and systemic lupus erythematosus, where their presence and titer correlate with disease activity and severity [14,15,18]. In our cohort, we report a prevalence of anti-hnRNP A2/B1 autoantibodies of 40%, higher than previously reported in SSc and comparable to findings in studies on SLE [13,16,17,19]. This increased prevalence may be attributed to the higher sensitivity and specificity of protein-IP compared to the immunoblot techniques traditionally employed [20].

Autoantibodies directed against components of the spliceosome complex have been extensively described in patients with SSc [21]. Among them, anti-U1RNP antibodies are the prototypical ones, being the most frequently and longest described, and are associated with a peculiar disease phenotype and distinct prognostic features [22]. More recently, protein immunoprecipitation has led to the identification of rarer autoantibody subsets targeting spliceosome proteins, although their putative clinical associations remain to be fully characterized [21]. Although not formally part of the spliceosome, hnRNPs are functionally closely related to the complex, regulating RNA maturation and splicing [23]. A pioneering study combining transcriptomic and proteomic analyses of skin samples has recently revealed distinctive abnormalities of alternative splicing and intron retention involving fibrosis-related genes, which were associated with SSc and correlated with fibrotic disease progression [24]. It remains to be clarified whether the detection of anti-hnRNP antibodies in SSc reflects broader alterations affecting nucleic acid metabolism and the function of the entire machinery sustaining transcription and protein translation, a topic underscored by the fact that most autoantibodies detected in connective tissue diseases are directed against such ribonucleoprotein complexes.

Earlier studies in SSc suggested a link between serum reactivity to hnRNP A2/B1 and the risk of developing arthritis [16,17], sometimes correlating with erosive disease features. We did not observe this association, possibly due to the limited sample size. Nevertheless, consistent with a prior study [16], we found a significant association between anti-hnRNP A2/B1 positivity and gastrointestinal involvement in SSc patients. Despite the sample size, this association was confirmed across different disease subsets, including subgroup analyses involving only patients with lcSSc or with concurrent ACA positivity. Therefore, it is unlikely that the observed association results from the spurious influence from other covariates as disease domains or serum autoantibodies. Given the paucity of reliable biomarkers for this domain, further research in this area is warranted.

The prevalence of autoantibodies against other hnRNP components has been scarcely explored in autoimmune diseases, and to the best of our knowledge, this is the first study examining a broad panel of hnRNP autoantibodies in SSc. Our results confirm the low reactivity of SSc sera against hnRNP components A1, C1/C3, and H. Hassfeld et al. previously reported anti-hnRNP antibodies against A2/B1 but no other subunits in a cohort including various autoimmune diseases, with only few SSc cases and no relevant findings specifically reported for this group [25]. In contrast, we observed the presence of anti-hnRNP L autoantibodies—present in 80% of patients—mostly showing a single protein-IP band, with a minority exhibiting two bands. Despite the small sample size, the band patterns differed between patients with and without a history of cancer. This finding is intriguing considering the involvement of hnRNP L in tumorigenesis, as shown by evidence of its upregulation in lung, prostate, bladder, and hepatocellular malignancies, correlating with tumoral proliferation, invasive capacity, and migration [26,27]. Moreover, high levels of autoantibodies to hnRNP L of the IgA isotype have been described in the sera of patients affected by ovarian cancer [28]. Therefore, further studies are necessary to determine whether anti-hnRNP L autoantibodies are characteristic of SSc, or rather represent a nonspecific autoimmune phenomenon potentially also present in healthy individuals or in the absence of autoimmune disease. Moreover, our preliminary observation on different anti-hnRNP L band patterns in SSc patients with or without a history of cancer requires validation in larger and independent cohorts.

Previous data from a Russian cohort suggested an association between anti-hnRNP B1 antibodies and digital ulcers in SSc [16]. While we did not replicate this finding—observing comparable ulcer prevalence in patients positive or negative for hnRNP A2/B1 antibodies—we identified one patient positive for anti-hnRNP U antibodies who presented with severe digital ulcers requiring multiple pharmacological interventions. As this is the first report of anti-hnRNP U positivity in SSc, this observation warrants further investigation.

Our study has several limitations that must be acknowledged. Due to the exploratory nature of the study and the small sample size, the statistical power and generalizability of the results remain limited, requiring validation in larger independent populations. This degree of uncertainty is reflected by the wide rage observed in the 95% CI of the ORs on multivariable logistic regression. Although a range of clinical manifestations was represented, the cohort was skewed towards a higher prevalence of limited cutaneous SSc and ACA positivity compared to larger European cohorts. On the one hand, this may have led to the inclusion of patients with less severe disease, potentially obscuring differences related to hnRNP status; on the other hand, we did not explore hnRNP subunits extensively in patients with more severe disease. Nevertheless, within our cohort, our observations remained consistent even in subgroup analyses, despite the small numbers. Our data derive from protein-IP assays and should be validated and compared with other methods, which—although generally less sensitive and specific—are more commonly used in routine clinical laboratories and therefore more readily applicable in clinical practice. Consequently, the results of this study must be interpreted cautiously and as preliminary and exploratory; validation in larger and more heterogeneous patient cohorts is necessary.

## 4. Materials and Methods

### 4.1. Patients, Sera, and Study Procedures

Serum samples were obtained from consecutive patients aged ≥ 18 years who fulfilled the 2013 classification criteria for SSc, with available follow-up at our Rheumatology Department for 10 years. Demographic, clinical, laboratory, and instrumental data were collected at the most recent follow-up visit. Disease domains were defined according to established reference standards and included LeRoy subsets (dcSSc, lcSSc, and *sine* scleroderma); interstitial lung disease (ILD—diagnosed by high-resolution chest CT scan) [29]; pulmonary arterial hypertension (PAH—diagnosed by right-heart catheterization) [30]; primary heart involvement (diagnosed by cardiac magnetic resonance imaging) [31]; and clinically defined digital ulcers, calcinosis, cutaneous telangiectasias, arthritis, myositis and gastrointestinal symptoms (i.e., at least one among gastroesophageal reflux, esophageal dysphagia or food impaction, refractory dyspepsia or early satiety, diarrhea, constipation, or malabsorption syndrome).

The study was approved by the Ethics Committee of IRCCS Humanitas Research Hospital, project code 831, approval date 20 April 2011. All procedures were conducted in accordance with the Declaration of Helsinki. Written informed consent was obtained from all participants.

### 4.2. Autoantibody Detection by Indirect Immunofluorescence (IFI), Protein-IP, and IP-Western Blot (WB)

Immunofluorescent antinuclear/cytoplasmic antibodies (HEp-2 antinuclear antibody (ANA) slides by INOVA Diagnostics, San Diego, CA, USA) were tested using serial dilutions of human sera (1:80, 1:160, 1:320, 1:640, and 1:1280) from patients and controls. Patients’ sera were incubated on HEp-2 slides followed by AlexaFluor488 AffiniPure F(ab’)2 fragment goat anti-human IgG (1:400 dilution, Jackson Immunoresearch Europe Ltd., Suffolk, UK). After incubation, the slides were washed with PBS and incubated with DAPI (1:20 dilution), and cover glasses were fixed on the slides using nail polish. Images were acquired immediately using an Olympus BX53 Upright fluorescence microscope. ANA patterns were defined according to the ICAP (International Consensus on Antinuclear antibody Patterns; https://anapatterns.org, accessed on 10 August 2025) classification [32].

Autoantibodies in the sera were screened by immunoprecipitation (IP) using ^35^S-methionine-labeled K562 cell extract [33]. Specificities of the autoantibodies were determined using reference sera.

Candidates for anti-hnRNP were selected based on the IP of a set of proteins of different molecular weights of the complex components, in particular: A1 (38 kDa), the two splicing variants A2 (34 kDa) and B1 (32 kDa), C1 (37 kDa) and C2 (39 kDa), H (50 kDa), L (64 KDa), and U (90 kDa). The identification of each protein of the hnRNP complex was verified by IP-Western blotting, as cell extract from 10^7^ K562 cells was immunoprecipitated using Protein A-Sepharose beads crosslinked with IgG from candidate sera [33]. Proteins were fractionated by 10% or 8% sodium dodecyl sulfate polyacrylamide gel electrophoresis (SDS-PAGE) depending on the molecular weight of the proteins and transferred to a nitrocellulose membrane. The membrane was probed with 2.4 ug/mL rabbit anti-A2B1, 0.25 ug/mL of rabbit anti-C1/C2, 0.025 ug/mL anti rabbit anti-H, or 0.25 ug/mL mouse anti-A1 and anti-U, followed by 1:5000 goat anti-rabbit IgG (Novus Biologicals, CO, USA) or 1:1000 goat anti-mouse IgG (Novus Biological, CO, USA). Signals from membranes were developed using Chemiluminescence detection Immobilon Western HRP substrate, and visualized using the ChemiDoc imaging system (Bio-Rad, CA, USA).

Due to antigen cross-reactivity, autoantibodies were considered to be directed against A2/B1 and C1/C2, respectively, and will be referred to accordingly hereafter, in line with previous observations [13].

### 4.3. Statistical Analysis

Continuous variables are expressed as means (±standard deviation) or medians (interquartile range), as appropriate; categorical variables are reported as numbers (percentages). Comparisons between continuous variables were performed using two-tailed *t*-tests or Mann−Whitney U tests, whereas categorical variables were compared using the chi-square or Fisher’s exact test, as appropriate.

Multivariable logistic regression was performed to evaluate covariates selected based on clinical relevance or a *p*-value < 0.10 in the univariable analysis. Given the sample size, the purely exploratory nature of the multivariable analysis is acknowledged.

A *p*-value < 0.05 was considered statistically significant. All analyses were performed using RStudio (v 2024.12.0+467).

## 5. Conclusions

This preliminary, hypothesis-generating study reports that anti-hnRNP autoantibodies are present in patients with SSc, with prevalence varying by component. Some components, such as U, are very rare and are reported here for the first time. Others, including A2/B1—previously characterized in other connective tissue diseases and rheumatoid arthritis—have intermediate prevalence and appear more frequent in patients with gastrointestinal involvement. Finally, considering the known link between hnRNPs and cancer, further studies are needed to clarify the clinical relevance of anti-hnRNP autoantibodies in the context of malignancy-associated autoimmunity.

## Figures and Tables

**Figure 1 ijms-26-08892-f001:**
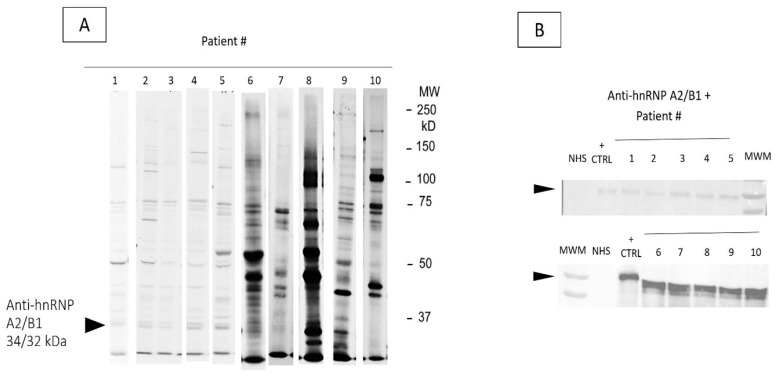
Detection of anti-hnRNP A2/B1 antibodies. (**A**) Protein-IP of anti-hnRNP A2/B1 autoantibodies (patient #1–10). The 35^S^-methionine labeled K562 cell extract was immunoprecipitated by human sera and analyzed by 8% SDS-PAGE. Anti-hnRNP A2/B1 sera immunoprecipitate 32–34 KDa proteins (indicated with the arrow for 10 patients), which are different from the mobility of other known autoantigens. (**B**) IP-Western blot of hnRNP A2/B1 positive sera. The identity of the 32–34 KDa proteins as hnRNP A2/B1 was verified by IP-Western. The protein is indicated with the arrow for 10 patients; the anti-hnRNP A2/B1 positive control (CTRL) is the commercial antibody immunoprecipitated as per protocol. NHS, normal human serum; MWM, molecular weight marker.

**Table 1 ijms-26-08892-t001:** Demographic, clinical, and laboratory characteristics of SSc patients enrolled in the study, according to the presence of anti-hnRNP A2/B1 autoantibodies. Sub-analyses conducted in the ACA-positive and lcSSc-only subgroups are also reported. Abbreviations: ACA, anticentromere antibody; dcSSc, diffuse cutaneous SSc; GI, gastrointestinal; ILD, interstitial lung disease; mRSS, modified Rodnan skin score; pHI, primary heart involvement; POLR3, RNA polymerase III; RHC-PAH, pulmonary arterial hypertension defined on right-heart catheterization; SRC, scleroderma renal crisis; TOPO1, topoisomerase I. Significant comparisons (*p* < 0.05) are reported with asterisks (*).

	Overall				ACA+			lcSSc		
	Overall (25)	hnRNP A2/B1+ (10)	hnRNP A2/B1− (15)	*p*	hnRNP A2/B1+ (6)	hnRNP A2/B1− (9)	*p*	hnRNP A2/B1+ (7)	hnRNP A2/B1− (14)	*p*
Age at onset	53 (14)	58 (10)	50 (15)	0.118	56 (11)	47 (16)	0.209	560 (10.61)	500 (15.10)	0.307
Duration	16 (10)	16.5 (9.5)	15 (9)	0.779	16 (10.5)	19 (8.5)	0.630	19.29 (9.79)	167 (9.38)	0.486
Sex	4 (16)	3 (30)	1 (7)	0.267	1 (17)	0 (0)	0.400	0 (0)	1 (7)	1.000
Smoke	8 (32)	2 (20)	6 (40)	0.402	1 (17)	2 (22)	1.000	0 (0)	5 (36)	0.124
Cancer	4 (16)	0 (0)	4 (27)	0.125	0 (0)	2 (22)	0.486	0 (0)	4 (29)	0.255
Alive	20 (80)	7 (70)	13 (87)	0.358	5 (83)	7 (78)	1.000	6 (86)	12 (86)	1.000
dcSSc	4 (16)	3 (30)	1 (7)	0.267	1 (17)	0 (0)	0.400	0 (0)	0 (0)	1.000
mRSS	4 (2–8)	6 (3–11)	3 (2–8)	0.186	4 (2–6)	3 (2–6)	0.669	4 (2.5–7.5)	3 (2–6.8)	0.423
ILD	5 (20)	1 (10)	4 (27)	0.615	0 (0)	1 (11)	1.000	0 (0)	3 (21)	0.521
pHI	5 (20)	3 (30)	2 (13)	0.358	1 (17)	1 (11)	1.000	2 (29)	1 (7)	0.247
RHC-PAH	2 (8)	1 (10)	1 (7)	1.000	1 (17)	1 (11)	1.000	1 (14)	1 (7)	1.000
GI involvement	12 (48)	8 (80)	4 (27)	**0.015 ***	6 (100)	3 (33)	**0.028 ***	6 (86)	3 (21)	**0.016 ***
Arthritis	5 (20)	2 (20)	3 (20)	1.000	0 (0)	1 (11)	1.000	1 (14)	2 (14)	1.000
SRC	0 (0)	0 (0)	0 (0)	0.317	0 (0)	0 (0)	0.439	0 (0)	0 (0)	0.127
Digital ulcers	11 (44)	4 (40)	7 (47)	1.000	3 (50)	6 (67)	0.622	3 (43)	6 (43)	1.000
Telangiectasias	15 (60)	5 (50)	10 (67)	0.677	4 (67)	7 (78)	1.000	5 (71)	10 (71)	1.000
Calcinosis	7 (28)	2 (20)	5 (33)	0.659	1 (17)	5 (56)	0.287	2 (29)	5 (36)	1.000
Myositis	0 (0)	0 (0)	0 (0)	0.317	0 (0)	0 (0)	0.439	0 (0)	0 (0)	0.127
ACA	15 (60)	6 (60)	9 (60)	1.000	6 (100)	9 (100)	1.000	5 (71)	9 (64)	1.000
anti-TOPO1	4 (16)	2 (20)	2 (13)	1.000	0 (0)	0 (0)	1.000	1 (14)	1 (7)	1.000
anti-POLR3	3 (12)	1 (10)	2 (13)	1.000	0 (0)	2 (22)	0.486	0 (0)	2 (14)	0.533
anti-SSA	5 (20)	2 (20)	3 (20)	1.000	1 (17)	1 (11)	1.000	1 (14)	2 (14)	1.000
anti-nucleolar	5 (25)	0 (0)	5 (33)	0.061	0 (0)	0 (0)	0.439	0 (0)	4 (29)	0.255

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
