# Peer review of "Detection and Clinical Associations of Autoantibodies to Heterogeneous Nuclear Ribonucleoprotein (hnRNP) A2/B1 in Patients with Systemic Sclerosis"

_ijms, 2025, doi:10.3390/ijms26188892_

Round 1
Reviewer 1 Report
Comments and Suggestions for Authors
The study by Tonutti A. et al aimed to detect autoantibodies to ribonucleoprotein A2B1 in the peripheral blood of systemic sclerosis patients. Although tу study is rather short, it presents new data related to antibody biology in autoimmunity. I did not find any significant flaws in the methodology or the results.
I have only two minor concerns to mention.
- The concluding statements, also mentioned in the abstract, state “Anti-hnRNP A2/B1 correlate with gastrointestinal involvement but, contrary to previous re-ports, show no association with arthritis. The potential link between anti-hnRNP L and cancer warrants further investigation” However, these conclusions are not directly related to the results obtained in this study. The conclusion must be revised to better reflect the main findings of the study.
- The authors should propose a possible mechanical molecular crosslink between autoantibody formation to hnRNP and the onset and progression of SS. Is there any connection to alternative splicing?
The article can be published after these minor corrections.
Author Response
The study by Tonutti A. et al aimed to detect autoantibodies to ribonucleoprotein A2B1 in the peripheral blood of systemic sclerosis patients. Although tу study is rather short, it presents new data related to antibody biology in autoimmunity. I did not find any significant flaws in the methodology or the results.I have only two minor concerns to mention.
- The concluding statements, also mentioned in the abstract, state “Anti-hnRNP A2/B1 correlate with gastrointestinal involvement but, contrary to previous re-ports, show no association with arthritis. The potential link between anti-hnRNP L and cancer warrants further investigation” However, these conclusions are not directly related to the results obtained in this study. The conclusion must be revised to better reflect the main findings of the study.
Response. We thank the Reviewer for this comment and in response we have added a revision of the conclusions in the Abstract conclusions in Page 2 and in the Conclusions section in Page 8.
- The authors should propose a possible mechanical molecular crosslink between autoantibody formation to hnRNP and the onset and progression of SS. Is there any connection to alternative splicing?
Response. We appreciate the Reviewer’s input on this point, and we have added a paragraph in the Discussion section at Page 5.
Reviewer 2 Report
Comments and Suggestions for Authors
This is a clinically relevant study that investigates addressing the prevalence and clinical associations of anti-hnRNP autoantibodies in systemic sclerosis (SSc). The authors revealed a correlation between anti-hnRNP A2/B1 and gastrointestinal involvement, which is noteworthy, as the biomarkers for gastrointestinal manifestations in SSc are limited. The manuscript is well-written. The introduction provides sufficient background on the topic. The results are appropriately presented. The references are adequate. However, there are several limitations of this study, including:
- The very small sample size (n=25) significantly limits the statistical power and generalizability of the results, as well as the soundness of the conclusions.
- The OR for gastrointestinal involvement has a very wide CI (2.2–381), further indicating the uncertainty of the OR due to the small sample.
- The authors should discuss whether there was any correlation between GI involvement and other clinical or serological features that might explain the link between anti-hnRNP A2/B1 and GI involvement.
- The observation regarding the band pattern of anti-hnRNP L and cancer history (p=0.066) is highly preliminary. This analysis is extremely fragile, with only four cancer cases in total. The discussion should include an elucidation of the established role of hnRNP L in tumorigenesis, with emphasis on the fact that this is a preliminary observation that requires validation in a much larger cohort.
- The abstract and conclusions should state that this is a preliminary, hypothesis-generating study.
Author Response
This is a clinically relevant study that investigates addressing the prevalence and clinical associations of anti-hnRNP autoantibodies in systemic sclerosis (SSc). The authors revealed a correlation between anti-hnRNP A2/B1 and gastrointestinal involvement, which is noteworthy, as the biomarkers for gastrointestinal manifestations in SSc are limited. The manuscript is well-written. The introduction provides sufficient background on the topic. The results are appropriately presented. The references are adequate. However, there are several limitations of this study, including:
- The very small sample size (n=25) significantly limits the statistical power and generalizability of the results, as well as the soundness of the conclusions.
Response: We agree with the Reviewer’s comment and we added a sentence on this limitation in the Discussion in Page 6.
- The OR for gastrointestinal involvement has a very wide CI (2.2–381), further indicating the uncertainty of the OR due to the small sample.
Response: As suggested by the Reviewer, we added a statement on this weak point in the Discussion section at Page 6.
- The authors should discuss whether there was any correlation between GI involvement and other clinical or serological features that might explain the link between anti-hnRNP A2/B1 and GI involvement.
Response: We appreciate the Reviewer’s input on this point and we have added a sentence in the Discussion at Page 6.
- The observation regarding the band pattern of anti-hnRNP L and cancer history (p=0.066) is highly preliminary. This analysis is extremely fragile, with only four cancer cases in total. The discussion should include an elucidation of the established role of hnRNP L in tumorigenesis, with emphasis on the fact that this is a preliminary observation that requires validation in a much larger cohort.
Response: We agree with the Reviewer’s point on this aspect and we have added a paragraph in the Discussion section at Page 6.
- The abstract and conclusions should state that this is a preliminary, hypothesis-generating study..
Response: We have followed the Reviewer’s suggestion and added one statement in the Abstract Conclusions in Page 2, in the Discussion in Page 6 and in the Conclusions section in Page 8.
Round 2
Reviewer 2 Report
Comments and Suggestions for Authors
I would like to thank the authors for considering my comments and for the amendments they made to the manuscript. The authors have provided a detailed point-by-point response and have made significant improvements to the manuscript, particularly in articulating the limitations associated with the small sample size.
The manuscript has been much improved, and the conclusions are now more appropriately cautious in interpreting the findings as hypothesis-generating. My major concerns have been addressed satisfactorily.